# Solving the Incompressible Surface Stokes Equation by Standard Velocity-Correction Projection Methods

**DOI:** 10.3390/e24101338

**Published:** 2022-09-23

**Authors:** Yanzi Zhao, Xinlong Feng

**Affiliations:** College of Mathematics and System Sciences, Xinjiang University, Urumqi 830017, China

**Keywords:** incompressible Stokes equation for surfaces, standard velocity correction projection method, mixed finite element pair, penalty term

## Abstract

In this paper, an effective numerical algorithm for the Stokes equation of a curved surface is presented and analyzed. The velocity field was decoupled from the pressure by the standard velocity correction projection method, and the penalty term was introduced to make the velocity satisfy the tangential condition. The first-order backward Euler scheme and second-order BDF scheme are used to discretize the time separately, and the stability of the two schemes is analyzed. The mixed finite element pair $\left(P_{2},P_{1}\right)$ is applied to discretization of space. Finally, numerical examples are given to verify the accuracy and effectiveness of the proposed method.

## 1. Introduction

Fluid equations on surfaces are used in mathematical modeling of emulsions [1], foams [2], and biofilms [3] where curved fluid membranes behave transversely as efficient viscoelastic media, as studied by Rahimi et al. [4]. Typically, such models consist of surface (Navier)–Stokes equations. These equations are also studied as an interesting mathematical problem in their own right. At present, the discretization methods of fluid equation on a curved surface are mainly the finite element method of a scalar elliptic equation and a parabolic partial differential Equation [4,5].

Due to the importance of surface incompressible flow in practical applications, many scholars have carried out in-depth research on it. Many achievements have also been made in the study of discretization methods of vector equations. Nitschke, Reuther et al. proposed the finite element method of Navier–Stokes equations of curved surfaces based on curl operators [6,7]. Reuther and Fries discretized the incompressible Navier–Stokes equations using a curved finite element method [8,9]. Reuther considered the $P_{1}-P_{1}$ finite element that does not satisfy inf-sup condition, and used the punishment technique to force the velocity field to be tangent to the surface. In contrast, Fries used the $P_{2}-P_{1}$ element, combined with the Lagrange multiplier method, to force the velocity to meet the tangential condition. However, the numerical analysis of the Navier–Stokes finite element method for curved surfaces was not involved in these two papers. Later, scholars began to use the finite element method to analyze the vector Laplace equation on the surface. This was the first step in extending the analysis of scalar problems to surface Navier–Stokes equations. They cross-analyzed a curved finite element method combined with a penalty technique to impose tangential constraints, and gave the results of stability analysis and error analysis, and also explained the influence of geometric errors on solving the problem [10]. A non-fitting finite element method, the trace finite element method, was proposed by Hansbo. When combining it with the Lagrange multiplier method, the forced velocity field satisfies the tangential condition, and the stability and optimal error estimation of the method are also proved [11].

The structure of this paper is as follows: The definition of surface differential operator is introduced in Section 2. In Section 3, the theory of surface finite element approximation and the Stokes equation are introduced. In Section 4, the standard format of first-order and second-order velocity correction methods is introduced and the stability analysis of semi-discrete time is proved. In Section 5, numerical examples are given to verify the accuracy and validity of the proposed method. The sixth part is the conclusion and prospects.

## 2. Surface Differential Operator

In this paper, we consider that the surface $\Gamma \subset R^{3}$ is closed and sufficiently smooth, and can be defined by a level set function satisfying ϕ(x)≠0, so that
$$\Gamma =\{x\left(x_{1},x_{2},x_{3}\right)\in R^{3}\;|\;\phi \left(x\right)=0\}.$$

The unit normal vector of the surface Γ at node *x* can be defined as
$$n\left(x\right)=\frac{\nabla \phi (x)}{\left|\nabla \phi (x)\right|}$$

On the surface Γ, the orthogonal projection $P\left(x\right)\in R^{3}$ is defined by the normal vector
$$P=P\left(x\right)=I-n\left(x\right)n{\left(x\right)}^{T}.$$

Define the neighborhood of the surface Γ as $U_{\delta }=\left\{x\in R^{3}\;|\;dist\left(x,\Gamma \right)\leq \delta \right\}.$ Thus, for any $x\in U_{\delta }$, there is the only point of a(x)∈Γ, such that
(1)$$x=a\left(x\right)+\tilde{d}\left(x\right)n\left(a\left(x\right)\right).$$

In addition, for the symbol distance function defined on the surface $\tilde{d}\left(x\right)=dist\left(x,\Gamma \right)$,
$$\nabla \tilde{d}\left(x\right)=n\left(a\left(x\right)\right),\left|\nabla \tilde{d}\left(x\right)\right|=1$$

The normal vector n is continuously extended along the normal direction in the neighborhood, and can be obtained by $n\left(x\right)=n\left(a\left(x\right)\right)=\nabla \tilde{d}\left(x\right).$

From the above property, the projection $P:U_{\delta }\to \Gamma$ of the nearest point in the neighborhood $U_{\delta }$ of the surface Γ is clearly defined. Thus, for a scalar function u(x), differentiable on the surface Γ, the tangential gradient operator $\nabla_{\Gamma }$ is defined as
$$\nabla_{\Gamma }u\left(x\right)=P\;\nabla \tilde{u},x\in \Gamma$$

Then, the gradient operator of a vector $u\left(x\right)={\left(u\left(x\right),v\left(x\right),w\left(x\right)\right)}^{T}$ is defined as follows:$$\nabla_{\Gamma }u\left(x\right)=P\;\nabla \tilde{u}\;P$$

Definition of the strain tensor of surface [12]:(2)$$E_{s}\left(u\right)=\frac{1}{2}P\left(\nabla u+\nabla u^{T}\right)P=\frac{1}{2}\left(\nabla_{\Gamma }u+\nabla_{\Gamma }u^{T}\right).$$

Similarly, for any vector u and tensor A, the surface divergence operator is defined as follows:$$div_{\Gamma }u:=tr\left(\nabla_{\Gamma }u\right)=tr\left(P\left(\nabla u\right)P\right)=tr\left(\left(\nabla u\right)P\right)=\nabla_{\Gamma }\times u,$$
$$div_{\Gamma }A:={\left(div_{\Gamma }\left(\varepsilon_{1}^{T}A\right),div_{\Gamma }\left(\varepsilon_{2}^{T}A\right),div_{\Gamma }\left(\varepsilon_{3}^{T}A\right)\right)}^{T}=\nabla_{\Gamma }\times A.$$
where: $\varepsilon_{i}$ (*i* = 1, 2, 3) is the i basis vector.

In this section, we review some surface differential operators of smooth functions and derive some properties of these operators. Therefore, tangent vector function space is introduced:$$C_{t}^{m}{\left(\Gamma \right)}^{3}:=\left\{u\in C^{m}{\left(\Gamma \right)}^{3}|on\Gamma ,n\cdot u=0\right\}.$$

Then, the curl operator can be defined as
$$rot_{\Gamma }u:=\left(\nabla_{\Gamma }\times u\right)\times n=div_{\Gamma }\left(u\times n\right),\;u\in C^{1}{\left(\Gamma \right)}^{3},$$
$${\mathrm{rot}}_{\Gamma }\psi :=n\times \nabla_{\Gamma }\psi ,\;\psi \in C^{1}\left(\Gamma \right).$$

In the Gaussian curvature κ(x) on the surface of Γ, for all the $\psi \in C^{1}\left(\Gamma \right)$, $u\in C_{t}^{1}{\left(\Gamma \right)}^{3}$, with the following formula:$$\begin{array}{cc} rot_{\Gamma }\left(\nabla_{\Gamma }\psi \right) & =0,\\ div_{\Gamma }\left({\mathrm{rot}}_{\Gamma }\psi \right) & =0,\\ rot_{\Gamma }\left({\mathrm{rot}}_{\Gamma }\psi \right) & =div_{\Gamma }\left(\nabla_{\Gamma }\psi \right),\\ {\mathrm{rot}}_{\Gamma }\left(rot_{\Gamma }u\right) & =Pdiv_{\Gamma }\left(\nabla_{\Gamma }u-\nabla_{\Gamma }u^{T}\right)=Pdiv_{\Gamma }\left(\nabla_{\Gamma }u\right)-\nabla_{\Gamma }\left(div_{\Gamma }u\right)-\kappa u. \end{array}$$

In the literature [13,14], the following equation is found:$$Pdiv_{\Gamma }\left(\nabla_{\Gamma }u^{T}\right)=\nabla_{\Gamma }\left(div_{\Gamma }u\right)+\kappa u.$$

When $u\in C_{t}^{2}{\left(\Gamma \right)}^{3}$ satisfies $div_{\Gamma }u=0$, then we have
$$Pdiv_{\Gamma }\left(2E_{s}\left(u\right)\right)={\mathrm{rot}}_{\Gamma }rot_{\Gamma }u+2\kappa u,$$
where: κ is Gaussian curvature.

**Remark** **1.**
*An internal form of the surface incompressible Stokes equation is independent of the selected coordinate system. Compared with the plane incompressible Stokes equation, not only is the corresponding operator replaced by the surface operator, but also, the Gaussian curvature has an additional contribution. This is because the surface strain tensor $E_{s}\left(u\right)=\frac{1}{2}(\nabla_{\Gamma }u$$+{\left(\nabla_{\Gamma }u\right)}^{T})$ divergence of the cause uses the Codazzi–Mainardi equation and the incompressible condition*

$$\begin{array}{cc} 2div_{\Gamma }\left(E_{s}\left(u\right)\right) & =div_{\Gamma }\nabla_{\Gamma }u+div_{\Gamma }\left(\nabla_{\Gamma }u^{T}\right)\\ \; & =-\Delta^{dR}u+\kappa u+\nabla_{\Gamma }\left(\nabla_{\Gamma }\times u\right)+\kappa u\\ \; & =-\Delta^{dR}u+2\kappa u. \end{array}$$

*where: $\Delta^{dR}u=-\left(\Delta^{RR}+\Delta^{GD}\right)u$, $\Delta^{RR}u={\mathrm{rot}}_{\Gamma }rot_{\Gamma }u$, $\Delta^{GD}u=\nabla_{\Gamma }\left(\nabla_{\Gamma }\times u\right)$, because of the incompressibility condition $\Delta^{dR}u={\mathrm{rot}}_{\Gamma }rot_{\Gamma }u$.*


## 3. Finite Element Approximation of Surface

In this section, the standard Hilbert space and some of the symbols reused in the following sections are introduced.
$$\begin{array}{cc} V & =\{v\in H^{1}{\left(\Gamma \right)}^{3}\},\\ Q & =\{q\in L^{2}{\left(\Gamma \right)}^{3}:\int_{\Gamma }qdx=0\},\\ X & =\{v\in H^{1}{\left(\Gamma \right)}^{3},\nabla_{\Gamma }\times v=0\}. \end{array}$$

There are a lot of studies about the numerical calculation of the two-dimensional Stokes equation in plane space [15,16,17], but there are few corresponding results on surfaces. The unsteady incompressible Stokes Equation [18] of a surface is shown below.
(3)$$\left\{\begin{array}{c} \partial_{t}u+\nabla_{\Gamma }p-\frac{1}{Re}Pdiv_{\Gamma }\left(E_{s}\left(u\right)\right)=f,\\ \nabla_{\Gamma }\times u=0. \end{array}\right.$$
where: *Re* denotes the surface Reynolds number, u∈V denotes the surface velocity, p∈Q denotes the surface pressure, $f\in L^{2}{\left(\Gamma \right)}^{3}$ denotes source term, and f×n=0. Pressure is constrained in the following form:(4)$$\begin{array}{c} \int_{\Gamma }\;p\;d\;x=0. \end{array}$$

In the Cartesian coordinate system, the curved Stokes equation is further modified by rotating the velocity field to reduce the complexity of the equation, and the original equation can be rewritten as
(5)$$\left\{\begin{array}{c} \partial_{t}u+\nabla_{\Gamma }p-\frac{1}{Re}\left({\mathrm{rot}}_{\Gamma }rot_{\Gamma }u+2\kappa u\right)+\alpha \left(u\times n\right)n=f,\\ \nabla_{\Gamma }\times u=0. \end{array}\right.$$

This formulation ensures the velocity to be tangential only weakly through the added penalty term and is equivalent to Equations (3) and (5) only if u×n=0 [19].

For weak formats of Stokes problems, spaces with norms are used to define $V:=H^{1}{\left(\Gamma \right)}^{3}$:$${\left||u|\right|}_{1}^{2}=\int_{\Gamma }{\left|u\left(x\right)\right|}^{2}+{\left|\nabla u\left(x\right)\right|}^{2}dx,$$

The corresponding space of the tangent vector field is defined as
$$V_{T}=\left\{u\in V\;|\;u\times n=0\;on\;\Gamma \right\}$$

For u∈V, the velocity field u is decomposed into tangential and normal components using the following symbols [20]:$$u=u_{T}+u_{N}n,\;u_{T}\times n=0.$$

Below, the time-dependent sequence’s finite difference is denoted by the following symbol [21]: {ϕ}.
(6)$$\begin{array}{cc} \delta \phi^{i+1}= & \phi^{i+1}-\phi^{i}, \end{array}$$
(7)$$\begin{array}{cc} \delta^{2}\phi^{i+1}= & \phi^{i+1}-2\phi^{i}+\phi^{i-1}, \end{array}$$
(8)$$\begin{array}{cc} D^{2}\phi^{i+1}= & 3\phi^{i+1}-4\phi^{i}+\phi^{i-1}. \end{array}$$

Finally, the following three identities are often used in theoretical analysis: (9)$$\begin{array}{cc} (a-b,2a)= & (a,a)-(b,b)+(a-b,a-b), \end{array}$$(10)$$\begin{array}{cc} (3a-4b+c,2a)= & \left(a,a)+(2a-b,2a-b)-(b,b)-(2b-c,2b-c\right)\\ \; & +(a-2b+c,a-2b+c), \end{array}$$(11)$$\begin{array}{cc} (3a-4b+c,2(a-b))= & \left(a-b,a-b)-(b-c,b-c\right)\\ \; & +(a-2b+c,a-2b+c)+4(a-b,a-b). \end{array}$$

## 4. Standard Velocity Correction Method

### 4.1. First-Order Velocity Correction Method

Firstly, the viscous velocity term is explicitly dealt with in the first step, and then it is modified in the second step. The corresponding scheme is as follows: ${\tilde{u}}^{0}=u^{0}$, and we select ${\tilde{u}}^{1}$ to better approximate u(Δt), when k≥1; solve $\left(u^{k+1},p^{k+1};{\tilde{u}}^{k+1}\right)$. Δt is the time step and has the following first-order format.

**Step 1.** We solve for $u^{k+1}\in V_{T},p^{k+1}\in Q$ from
(12)$$\left\{\begin{array}{c} \frac{u^{k+1}-{\tilde{u}}^{k}}{\Delta t}-\frac{1}{Re}\left({\mathrm{rot}}_{\Gamma }rot_{\Gamma }{\tilde{u}}^{k}+2\kappa {\tilde{u}}^{k}\right)+\alpha \left({\tilde{u}}^{k+1}\times n\right)n+\nabla_{\Gamma }p^{k+1}=f^{k+1},\\ \nabla_{\Gamma }\times u^{k+1}=0, \end{array}\right.$$

**Step 2**. We solve for ${\tilde{u}}^{k+1}\in V_{T}$ from
(13)$$\frac{{\tilde{u}}^{k+1}-u^{k+1}}{\Delta t}-\left({\mathrm{rot}}_{\Gamma }rot_{\Gamma }\left({\tilde{u}}^{k+1}-{\tilde{u}}^{k}\right)+2\kappa \left({\tilde{u}}^{k+1}-{\tilde{u}}^{k}\right)\right)=0.$$**Implementation of the standard form**


It is difficult to solve Equation (Equation 12) as a weak Poisson problem for pressure due to the existence of the second derivative of Equation (Equation 12). To avoid this difficulty, the algorithm can be rewritten into an equivalent form by algebraic substitution.

By subtracting (Equation 12) of *k* step format from (Equation 12) of k+1 step format and by substituting (Equation 13) of *k* step format into the resulting equation, a more adequate form of the projection step (Equation 14) is obtained. The actual implementation steps are as follows.

**Step 1**. We solve for $u^{k+1}\in V_{T},p^{k+1}\in Q$ from
(14)$$\left\{\begin{array}{c} \frac{u^{k+1}-2{\tilde{u}}^{k}+{\tilde{u}}^{k-1}}{\Delta t}+\nabla_{\Gamma }\left(p^{k+1}-p^{k}\right)=f^{k+1}-f^{k},\\ \nabla_{\Gamma }\times u^{k+1}=0, \end{array}\right.$$
Equation (Equation 14) can be rewritten as
(15)$$\frac{1}{\Delta t}\nabla_{\Gamma }\times \left(-2{\tilde{u}}^{k}+{\tilde{u}}^{k-1}\right)+\Delta_{\Gamma }\left(p^{k+1}-p^{k}\right)=\nabla_{\Gamma }\times \left(f^{k+1}-f^{k}\right),$$

**Step 2**. We solve for ${\tilde{u}}^{k+1}\in V_{T}$ from
(16)$$\frac{{\tilde{u}}^{k+1}-{\tilde{u}}^{k}}{\Delta t}-\frac{1}{Re}\left({\mathrm{rot}}_{\Gamma }rot_{\Gamma }{\tilde{u}}^{k+1}+2\kappa {\tilde{u}}^{k+1}\right)+\alpha \left({\tilde{u}}^{k+1}\times n\right)n=f^{k+1}-\nabla_{\Gamma }p^{k+1}.$$
Note that in algorithm (Equation 14)–(Equation 16), projection velocity $u^{k+1}$ be completely eliminated. Therefore, in the actual calculation, consider using approximate velocity ${\tilde{u}}^{k+1}$ instead of $u^{k+1}$.

### 4.2. Second-Order Velocity Correction Method

Similarly, the time semi-discrete scheme of the second-order standard velocity correction method is given below. ${\tilde{u}}^{0}=u^{0}$ and select ${\tilde{u}}^{1}$ to better approximate u(Δt); when k≥1, solve $\left(u^{k+1},p^{k+1};{\tilde{u}}^{k+1}\right)$; Δt is the time step with the following second-order format.

**Step 1**. We solve for $u^{k+1}\in V_{T},p^{k+1}\in Q$ from
(17)$$\left\{\begin{array}{c} \frac{1}{2\Delta t}\left(3u^{k+1}-4{\tilde{u}}^{k}+{\tilde{u}}^{k-1}\right)-\frac{1}{Re}\left({\mathrm{rot}}_{\Gamma }rot_{\Gamma }{\tilde{u}}^{k}+2\kappa {\tilde{u}}^{k}\right)+\alpha \left({\tilde{u}}^{k+1}\times n\right)n+\nabla_{\Gamma }p^{k+1}=f^{k+1},\\ \nabla_{\Gamma }\times u^{k+1}=0, \end{array}\right.$$

**Step 2**. We solve for ${\tilde{u}}^{k+1}\in V_{T}$ from
(18)$$\frac{3}{2\Delta t}\left({\tilde{u}}^{k+1}-u^{k+1}\right)-\left({\mathrm{rot}}_{\Gamma }rot_{\Gamma }\left({\tilde{u}}^{k+1}-{\tilde{u}}^{k}\right)+2\kappa \left({\tilde{u}}^{k+1}-{\tilde{u}}^{k}\right)\right)=0.$$**Implementation of the standard form**


It is difficult to solve Equation (Equation 17) as a weak Poisson problem for pressure due to the existence of the second derivative of the equation. To avoid this difficulty, the algorithm can be rewritten into an equivalent form by algebraic substitution.

Through (Equation 17) of k+1, step format minus *k*, step format, and combining (Equation 18) of *k* step format, we get a new step projection of (Equation 19) type. The actual implementation steps are as follows.

**Step 1**. We solve for $u^{k+1}\in V_{T},p^{k+1}\in Q$ from
(19)$$\left\{\begin{array}{c} \frac{1}{2\Delta t}\left(3u^{k+1}-7{\tilde{u}}^{k}+5{\tilde{u}}^{k-1}-{\tilde{u}}^{k-2}\right)+\nabla_{\Gamma }\left(p^{k+1}-p^{k}\right)=f^{k+1}-f^{k},\\ \nabla_{\Gamma }\times u^{k+1}=0, \end{array}\right.$$
Equation (Equation 19) can be rewritten as
(20)$$\frac{1}{2\Delta t}\nabla_{\Gamma }\times \left(-7{\tilde{u}}^{k}+5{\tilde{u}}^{k-1}-{\tilde{u}}^{k-2}\right)+\Delta_{\Gamma }\left(p^{k+1}-p^{k}\right)=\nabla_{\Gamma }\times \left(f^{k+1}-f^{k}\right),$$

**Step 2**. We solve ${\tilde{u}}^{k+1}\in V_{T}$ from
(21)$$\begin{array}{c} \begin{array}{cc} \frac{1}{2\Delta t}\left(3{\tilde{u}}^{k+1}-4{\tilde{u}}^{k}+{\tilde{u}}^{k-1}\right)-\frac{1}{Re}\left({\mathrm{rot}}_{\Gamma }rot_{\Gamma }{\tilde{u}}^{k+1}+2\kappa {\tilde{u}}^{k+1}\right)+\alpha \left({\tilde{u}}^{k+1}\times n\right)n\\ & =f^{k+1}-\nabla_{\Gamma }p^{k+1}. \end{array} \end{array}$$
Note once again that projection velocity $u^{k+1}$ be completely eliminated from the algorithm (Equation 19)–(Equation 21). Therefore, in the actual calculation consider using approximate velocity ${\tilde{u}}^{k+1}$ instead of $u^{k+1}$.

### 4.3. Stability Analysis

In this section, the stability analysis of the first and second-order time semi-discrete schemes of the standard velocity correction projection method is established. Without loss of generality, we assume f≡0.

#### 4.3.1. First-Order Scheme Stability Analysis

A key step in establishing the stability analysis results is to reformulate the standard velocity correction scheme using the Gauge–Uzawa formula. To be more precise, we introduce the auxiliary variable $w^{k}$ and define it as follows.
(22)$$w^{k}=-Pdiv_{\Gamma }E_{s}\left({\tilde{u}}^{k}\right),$$
Equation (Equation 13) can be rewritten as
(23)$${\tilde{u}}^{k+1}+\Delta tw^{k+1}=u^{k+1}+\Delta tw^{k}.$$

**Theorem** **1.**
*Equations (Equation 12) and (Equation 13) in f≡0 are unconditionally stable. Under the condition of time in each layer of 0≤k≤T/Δt−1, we have*

$$\varepsilon_{k+1}^{\left(1\right)}-\varepsilon_{k}^{\left(1\right)}\leq -\Delta t||E_{s}\left({\tilde{u}}^{k+1}\right){\left|\right|}^{2},$$

*where:*

$$\varepsilon_{k}^{\left(1\right)}=\left|\right|{\tilde{u}}^{k}{\left|\right|}^{2}+\Delta t||w^{k}{\left|\right|}^{2},$$

*is the corrected energy with time step k.*


**Proof.** Taking the inner product of $2\Delta tu^{k+1}$ with Equation (Equation 12), we obtain
(24)$$\left|\right|u^{k+1}{\left|\right|}^{2}-\left|\right|{\tilde{u}}^{k}{\left|\right|}^{2}+\left|\right|u^{k+1}-{\tilde{u}}^{k}{\left|\right|}^{2}-2\Delta t\left(Pdiv_{\Gamma }E_{s}\left(u^{k}\right),u^{k+1}\right)=0,$$The inner product of both sides of Equation (Equation 23) with itself:
(25)$$\left|\right|{\tilde{u}}^{k+1}{\left|\right|}^{2}+\Delta t^{2}\left|\right|w^{k+1}\left|\right|+2\Delta t\left({\tilde{u}}^{k+1}w^{k+1}\right)=\left|\right|u^{k+1}{\left|\right|}^{2}+\Delta t^{2}\left|\right|w^{k}{\left|\right|}^{2}+2\Delta t\left(u^{k+1},w^{k}\right),$$Readily available is
(26)$$\left({\tilde{u}}^{k+1},w^{k+1}\right)=\left|\right|E_{s}\left({\tilde{u}}^{k+1}\right){\left|\right|}^{2},$$
(27)$$\left(u^{k+1},w^{k}\right)=-\left(u^{k+1},Pdiv_{\Gamma }E_{s}\left({\tilde{u}}^{k}\right)\right).$$Substitute Equations (Equation 26) and (Equation 27) into Equation (Equation 25), and combine with Equation (Equation 24) to get the result
(28)$$\left|\right|{\tilde{u}}^{k+1}{\left|\right|}^{2}-\left|\right|{\tilde{u}}^{k}{\left|\right|}^{2}+\left|\right|u^{k+1}-{\tilde{u}}^{k}{\left|\right|}^{2}+\Delta t^{2}\left(|\right|w^{k+1}{\left|\right|}^{2}-\left|\right|w^{k}{\left|\right|}^{2})-2\Delta t||E_{s}{\left(\tilde{u}\right)}^{k+1}{\left|\right|}^{2}=0.$$□

#### 4.3.2. Stability Analysis of Second-Order Schemes

As with the first-order proof stability analysis method, an auxiliary variable $w^{k}$ is introduced and defined as follows:(29)$$w^{k}=-Pdiv_{\Gamma }E_{s}\left({\tilde{u}}^{k}\right),$$
We can rewrite (Equation 18) equation as
(30)$$3{\tilde{u}}^{k+1}+2\Delta tw^{k+1}=3u^{k+1}+2\Delta tw^{k}.$$

**Theorem** **2.**
*Equations (Equation 17) and (Equation 18) represent unconditional energy stability. For all 0≤k≤T/Δt−1, there is*

(31)
$$\begin{array}{c} \varepsilon_{k+1}^{\left(2\right)}-\varepsilon_{k}^{\left(2\right)}\leq -\Delta t||\nabla {\tilde{u}}^{k+1}{\left|\right|}^{2}, \end{array}$$

*where:*

$$\begin{array}{c} \varepsilon_{k}^{\left(2\right)}=\left|\right|{\tilde{u}}^{k}{\left|\right|}^{2}+||2{\tilde{u}}^{k}-{\tilde{u}}^{k-1}{\left|\right|}^{2}+\frac{2\Delta }{3}\left|\right|E_{s}\delta \left({\tilde{u}}^{k}\right){\left|\right|}^{2}+\frac{4\Delta t^{2}}{3}\left|\right|w^{k}{\left|\right|}^{2}, \end{array}$$

*is the correction energy of the time step k.*


**Proof.** Take the inner product of (Equation 17) with $4\Delta tu^{k+1}$, to get
(32)$$\begin{array}{c} I+(-Pdiv_{\Gamma }E_{s}{\left(\tilde{u}\right)}^{k+1},4\Delta {\tilde{u}}^{k+1})=0, \end{array}$$
where:
$$\begin{array}{c} I=(3u^{k+1}-4{\tilde{u}}^{k}+{\tilde{u}}^{k-1},2u^{k+1}). \end{array}$$We deal with this term by using a similar treatment as in [22]. It can be rewritten *I* as
$$\begin{array}{c} I=2\left(D^{2}{\tilde{u}}^{k+1},{\tilde{u}}^{k+1}\right)+2\left(D^{2}{\tilde{u}}^{k+1},u^{k+1}-{\tilde{u}}^{k+1}\right)+6\left(u^{k+1}-{\tilde{u}}^{k+1},u^{k+1}\right):=I_{1}+I_{2}+I_{3}. \end{array}$$Using Equations (7) and (10), we can rewrite $I_{1}$ as
(33)$$\begin{array}{c} I_{1}=\left|\right|{\tilde{u}}^{k+1}{\left|\right|}^{2}+||2{\tilde{u}}^{k+1}-{\tilde{u}}^{k}{\left|\right|}^{2}-\left|\right|{\tilde{u}}^{k}{\left|\right|}^{2}-||2{\tilde{u}}^{k}-{\tilde{u}}^{k-1}{\left|\right|}^{2}+\left|\right|\delta^{2}{\tilde{u}}^{k+1}{\left|\right|}^{2}. \end{array}$$Thanks to Equations (Equation 29) and (Equation 30), we can rewrite $I_{2}$ as
(34)$$\begin{array}{cc} I_{2}\\ =\frac{4\Delta t}{3}\left(D^{2}{\tilde{u}}^{k+1},w^{k+1}-w^{k}\right)\\ \; & =\frac{2\Delta t}{3}\left(D^{2}{\tilde{u}}^{k+1},2\left(Pdiv_{\Gamma }E_{s}\left({\tilde{u}}^{k+1}\right)-Pdiv_{\Gamma }E_{s}\left({\tilde{u}}^{k}\right)\right)\right), \end{array}$$By Equations (Equation 6), (7) and (11), we can obtain
(35)$$\begin{array}{c} I_{2}=\frac{2\Delta t}{3}\left(|\right|E_{s}\delta \left({\tilde{u}}^{k+1}\right){\left|\right|}^{2}-\left|\right|E_{s}\delta \left({\tilde{u}}^{k}\right){\left|\right|}^{2}+\left|\right|E_{s}\delta^{2}\left({\tilde{u}}^{k+1}\right){\left|\right|}^{2}+4||E_{s}\delta \left({\tilde{u}}^{k+1}\right){\left|\right|}^{2}), \end{array}$$Equation (Equation 9) tells us that
(36)$$\begin{array}{c} I_{3}=3(||u^{k+1}{\left|\right|}^{2}-\left|\right|{\tilde{u}}^{k+1}{\left|\right|}^{2}+\left|\right|u^{k+1}-{\tilde{u}}^{k+1}{\left|\right|}^{2}). \end{array}$$Next, taking the inner product of both sides of Equation (Equation 30) with itself, we get
(37)$$\begin{array}{c} 3||{\tilde{u}}^{k+1}{\left|\right|}^{2}+\frac{4\Delta t^{2}}{3}\left|\right|w^{k+1}\left|\right|+4\Delta t\left({\tilde{u}}^{k+1},w^{k+1}\right)=3||u^{k+1}{\left|\right|}^{2}+\frac{4\Delta t^{2}}{3}\left|\right|w^{k}\left|\right|+4\Delta t\left(u^{k+1},w^{k}\right), \end{array}$$Equation (Equation 37) can also be written as Equation (Equation 38)
(38)$$\begin{array}{c} 3(||{\tilde{u}}^{k+1}{\left|\right|}^{2}-3||u^{k+1}{\left|\right|}^{2})+\frac{4\Delta t}{3}\left(|\right|w^{k+1}{\left|\right|}^{2}-\left|\right|w^{k}{\left|\right|}^{2})+4\Delta t\left({\tilde{u}}^{k+1},w^{k+1}\right)\\ =4\Delta t\left(u^{k+1},w^{k}\right)=4\Delta t\left(u^{k+1},-Pdiv_{\Gamma }E_{s}\left({\tilde{u}}^{k}\right)\right), \end{array}$$Summing up (Equation 32) and (Equation 38), and taking into account (Equation 33), (Equation 35) and (Equation 36), we can obtain
$$\begin{array}{c} \left|\right|{\tilde{u}}^{k+1}{\left|\right|}^{2}+||2{\tilde{u}}^{k+1}-{\tilde{u}}^{k}{\left|\right|}^{2}-\left|\right|{\tilde{u}}^{k}{\left|\right|}^{2}-||2{\tilde{u}}^{k}-{\tilde{u}}^{k-1}{\left|\right|}^{2}+\frac{2\Delta t}{3}\left(|\right|E_{s}\delta {\left(\tilde{u}\right)}^{k+1}{\left|\right|}^{2}-\left|\right|E_{s}\delta \left({\tilde{u}}^{k}\right){\left|\right|}^{2}\\ +\left|\right|E_{s}\delta^{2}\left({\tilde{u}}^{k+1}\right){\left|\right|}^{2}+4||E_{s}\delta \left({\tilde{u}}^{k+1}\right)\left||\right)+3(||{\tilde{u}}^{k+1}{\left|\right|}^{2}-\left|\right|{\tilde{u}}^{k+1}{\left|\right|}^{2}+||u-{\tilde{u}}^{k+1}{\left|\right|}^{2})\\ +3(||{\tilde{u}}^{k+1}{\left|\right|}^{2}-\left|\right|u^{k+1}{\left|\right|}^{2})+\frac{4\Delta t^{2}}{3}\left(|\right|w^{k+1}{\left|\right|}^{2}-\left|\right|w^{k}{\left|\right|}^{2})+4\Delta t\left({\tilde{u}}^{k+1},w^{k+1}\right)=0. \end{array}$$□

### 4.4. Fully Discrete Scheme

The first-order backward Euler and second-order BDF are used in discrete-time, and the mixed finite element pair is used in discrete space. $\Gamma_{h}=\left\{K\right\}$ is Γ on a consistent regular triangular mesh, and the grid size is $h=max_{K\in \Gamma_{h}}\left\{diam\left\{K\right\}\right\}$. Define the following discrete subspace $\left(V_{Th},Q_{h}\right)\subset \left(V_{T},Q\right)$:$$\begin{array}{c} V_{Th}=\left\{v_{h}\in V_{T}:v_{h}{|}_{K}\in {\left[P_{2}\left(K\right)\right]}^{2},\forall K\in \Gamma_{h}\right\},\\ Q_{h}=\left\{q_{h}\in Q:q_{h}{|}_{K}\in P_{1}\left(K\right),\forall K\in \Gamma_{h}\right\}. \end{array}$$

#### 4.4.1. First-Order Fully Discrete Velocity Correction Method

**Step 1**. For $\forall v_{ih}\in V_{Th},\forall q_{h}\in Q_{h}$, we solve for $\left(u_{h}^{k+1},p_{h}^{k+1}\right)$ from
(39)$$-\int_{\Gamma h}\left(-2{\tilde{u}}_{h}^{k}+{\tilde{u}}_{h}^{k+1}\right)\times \nabla_{\Gamma }q_{h}dx-\int_{\Gamma h}\left(p_{h}^{k+1}-p_{h}^{k}\right)\times \nabla_{\Gamma }q_{h}dx=-\int_{\Gamma h}\left(f^{k+1}-f^{k}\right)\times \nabla_{\Gamma }q_{h}dx,$$

**Step 2**. For $\forall v_{ih}\in V_{T}$, we solve for ${\tilde{u}}_{h}^{k+1}\in V_{Th}$ from
(40)$$\begin{array}{c} \begin{array}{cc} & \frac{1}{\Delta t}\int_{\Gamma h}\left({\tilde{u}}_{ih}^{k+1}-{\tilde{u}}_{ih}^{k}\right)v_{ih}dx-\frac{1}{Re}\int_{\Gamma h}rot_{\Gamma }{\tilde{u}}_{h}^{k+1}rot_{\Gamma }\left(v_{ih}e^{i}\right)dx-2\int_{\Gamma h}\kappa {\tilde{u}}_{ih}^{k+1}v_{ih}dx\\ & +\alpha \int_{\Gamma h}n\times {\tilde{u}}_{h}^{k+1}n_{i}v_{ih}dx=\int_{\Gamma h}f^{k+1}v_{ih}dx-\int_{\Gamma h}\nabla_{\Gamma }p_{h}^{k+1}v_{ih}dx. \end{array} \end{array}$$

#### 4.4.2. Second-Order Fully Discrete Velocity Correction Method

**Step 1**. For $\forall v_{ih}\in V_{Th},\forall q_{h}\in Q_{h}$, we solve for $\left(u_{h}^{k+1},p_{h}^{k+1}\right)$ from
(41)$$\begin{array}{c} \begin{array}{cc} & -\frac{1}{2\Delta t}\int_{\Gamma h}\left(-7{\tilde{u}}_{h}^{k}+5{\tilde{u}}_{h}^{k-1}-{\tilde{u}}_{h}^{k-2}\right)\times \nabla_{\Gamma }q_{h}dx-\int_{\Gamma h}\left(p_{h}^{k+1}-p_{h}^{k}\right)\times \nabla_{\Gamma }q_{h})dx\\ & =-\int_{\Gamma h}\left(f^{k+1}-f^{k}\right)\times \nabla_{\Gamma }q_{h}dx, \end{array} \end{array}$$

**Step 2**. For $\forall v_{ih}\in V_{T}$, we solve for ${\tilde{u}}_{h}^{k+1}\in V_{Th}$ from
(42)$$\begin{array}{c} \begin{array}{cc} & \frac{1}{2\Delta t}\int_{\Gamma h}\left(3{\tilde{u}}_{ih}^{k+1}-4{\tilde{u}}_{ih}^{k}+{\tilde{u}}_{ih}^{k-1}\right)v_{ih}dx-\frac{1}{Re}\int_{\Gamma h}rot_{\Gamma }{\tilde{u}}^{k+1}rot_{\Gamma }\left(v_{ih}e^{i}\right)dx-2\int_{\Gamma h}\kappa {\tilde{u}}_{ih}^{k+1}v_{ih}dx\\ & +\alpha \int_{\Gamma h}n\times {\tilde{u}}_{h}^{k+1}n_{i}v_{ih}dx=\int_{\Gamma h}f^{k+1}v_{ih}-\int_{\Gamma h}\nabla_{\Gamma }p_{h}^{k+1}v_{ih}dx. \end{array} \end{array}$$

## 5. Numerical Experiments

In this section, some numerical examples are designed to verify the effectiveness and accuracy of the velocity correction method; test the convergence and stability; and simulate the physical phenomena of circulating flow in a single fluid system.

### 5.1. Convergence Test

In order to verify the accuracy and effectiveness of this method, two different surfaces $\phi_{1}$ and $\phi_{2}$ were selected for a convergence test.

The spherical implicit function:$$\phi_{1}\left(x,y,z\right)=x^{2}+y^{2}+z^{2}-1.$$

The ring implicit function:$$\phi_{2}\left(x,y,z\right)={\left(\sqrt{x^{2}+z^{2}}-2\right)}^{2}+y^{2}-0.25=0.$$

The exact solution of the given surface Stokes problem is as follows.
$$\left\{\begin{array}{c} u\left(x,y,z,t\right)={\left(ye^{-t},-xe^{-t},0\right)}^{T},\\ p\left(x,y,z,t\right)=\left(xy^{3}+z\right)e^{-t}. \end{array}\right.$$

For the first-order numerical discrete scheme, let Re=10, t=0.1, $\alpha =1000/h^{2}$, time step for $\Delta t=h^{2}$, h=0.25,0.15,0.125,0.06,0.03. The order of convergence is calculated by the following formula [23]:(43)$$rate=\log\left(e_{i}/e_{i+1}\right)/\log\left(h_{i}/h_{i+1}\right).$$

Figure 1 shows convergence on a sphere and a ring with different surfaces of the first-order scheme. The results show that pressure *p* is in the $L^{2}$ norm of convergence in order to achieve second-order convergence, and velocity u is in the $L^{2}$ norm under second-order convergence. This is because of the existence of spacial geometric error.

### 5.2. A Circular Flow on a Ring

In this experiment, the unsteady curvature surface $\phi_{2}$ is used to simulate the physical phenomenon of the flow velocity change of the fluid on the ring [24,25].

Set the initial value $u_{0}$ as follows.
$$u_{0}={\left(\frac{2xy+z}{8(x^{2}+z^{2})},\frac{x^{2}+z^{2}-2\sqrt{x^{2}+z^{2}}}{4(x^{2}+z^{2})},\frac{-2y^{2}+x}{8(x^{2}+z^{3})}\right)}^{T}$$

Take Re=10, h=0.3, α=3000, and Δt=0.1 (Figure 2) to simulate the single fluid system. The initial velocity $u_{0}$ will gradually evolve into periodic flow over time. When the penalty coefficient α=0, the velocity field is not tangent to the surface, and periodic flow cannot be generated. Simulation results show that the proposed method can effectively simulate the physical phenomena of fluid flow and produce the expected intra-ring circulating flow.

### 5.3. Stability Test

The system is studied when the external force f=0. Consider a sphere with constant curvature $\phi_{1}\left(x,y,z\right)=x^{2}+y^{2}+z^{2}-1.$

Select initial conditions:$$\left\{\begin{array}{c} u_{0}\left(x,y,z\right)=2\cos\left(\pi y\right)\sin\left(\pi x\right)\cos\left(\pi z\right),\\ v_{0}\left(x,y,z\right)=-\sin\left(\pi y\right)\cos\left(\pi x\right)\cos\left(\pi z\right),\\ w_{0}\left(x,y,z\right)=-\cos\left(\pi y\right)\sin\left(\pi z\right)\cos\left(\pi x\right) \end{array}\right.$$

Let Re=10, t=1, $h=\frac{1}{60}$, and $\alpha =1000/h^{2}$. The energy dissipation diagrams of the first- and second-order velocity correction projection methods are given, respectively.

The system energy $E_{n}:=\left|\right|u^{n}{\left|\right|}^{2}+||vs.^{n}{\left|\right|}^{2}+\left|\right|w^{n}{\left|\right|}^{2}$ in Figure 3 shows the system in each time step: Δt=0.1,0.05,0.025,0.0125. The results show that the energy curve decreases to 0, which indicates that the energy is dissipated and verifies the stability of the algorithm.

## 6. Conclusions

In the curved Stokes fluid model, the velocity and pressure are decoupled by the standard velocity correction method, and the equivalent elliptic partial differential equation is obtained. The first-order and second-order numerical schemes of $\left(P_{2},P_{1}\right)$ are constructed by using curved surface mixed finite element matching for space discretization. The stability analysis of first-order and second-order time semi-discretization is established. Finally, three examples are given to verify the accuracy and effectiveness of the proposed method.

## Figures and Tables

**Figure 1 entropy-24-01338-f001:**
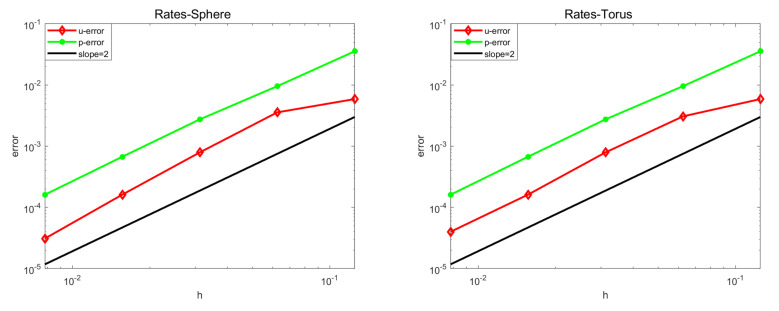
Orders of convergence of sphere (**left**) and ring (**right**).

**Figure 2 entropy-24-01338-f002:**
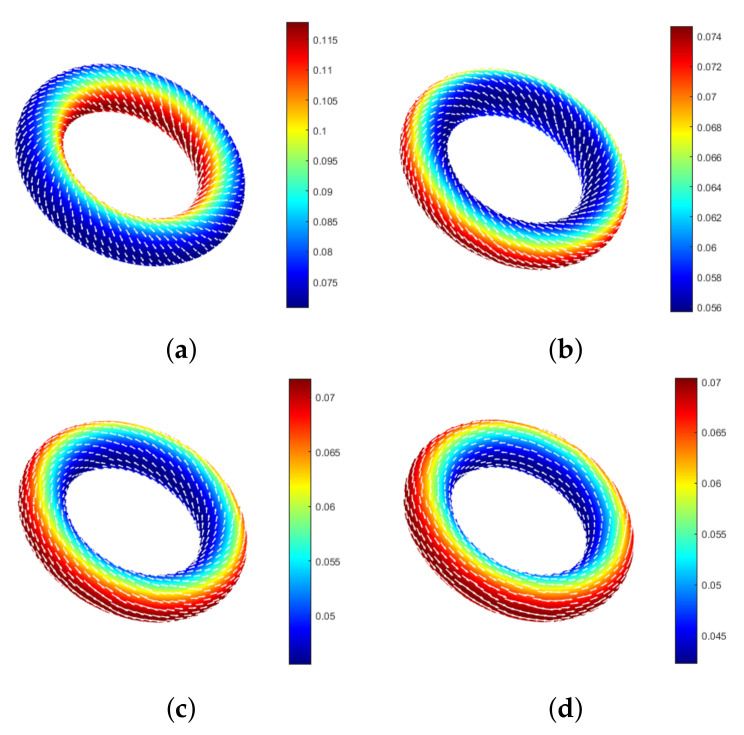
Flow diagram on the ring. (**a**) t=0. (**b**) t=15. (**c**) t=30. (**d**) t=90.

**Figure 3 entropy-24-01338-f003:**
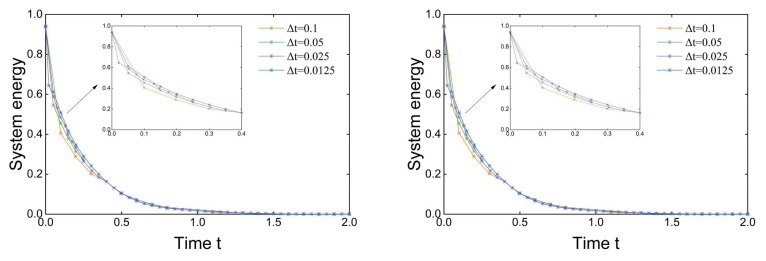
Energy dissipation of first-order (**left**) and second-order (**right**) systems.

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
