# Peer review of "Solving the Incompressible Surface Stokes Equation by Standard Velocity-Correction Projection Methods"

_entropy, 2022, doi:10.3390/e24101338_

Round 1
Reviewer 1 Report
see the attached file

Author Response
1

Reviewer 2 Report
This study is mainly focused on the mathematical formulation of the Stokes equation on a surface. Generally, this is a pure mathematical study, and not enough fluid flow analyses are provided. This study is more suitable for mathematical analysis journals.
Author Response
2

Reviewer 3 Report
Title: Standard velocity-correction projection methods for Stokes equation on surfaces
Manuscript ID: Entropy-1852263
Report: In this work, the numerical algorithm for the Stokes equation of curved surface is presented and analyzed. The velocity field was decoupled from the pressure by the standard velocity correction projection method, and the penalty term was introduced to make the velocity satisfy the tangential condition. The first order backwards Euler scheme and second order BDF scheme were used to discretize the time, respectively, and the stability of the two schemes is analyzed. The mixed finite element pair (P2, P1) is applied to space discretization. Finally, numerical examples were given to verify the accuracy and effectiveness of the proposed method. The stability analysis of first order and second order time semi-discretization is developed.
The paper is very well written. An effective numerical algorithm is established.
This paper may be accepted for publication in Entropy Journal’s special issue
On “Finite Element Methods for the Navier-Stokes Equations and MHD Equations”. However, the English language may be corrected.
Author Response
3

Round 2
Reviewer 1 Report
The authors have tried to answer to my comments, and I am satisfied with the answers,
the only remaining problem is the very bad quality of the pictures.
the author's answer is "we don't know how to improve them", but if they made the paper, they must know how to generate the figures, and therefore to improve them. Incomprehensible !!!
It is the problem of reputations, of the authors and of the journal. It is not my problem.
Then you can proceed to publication if you want.
Author Response
1
